# Outcomes of primary rhegmatogenous retinal detachment repair with extensive scleral-depressed vitreous removal and dynamic examination

Tedi Begaj[1☯], Anna Marmalidou[1☯], Thanos D. Papakostas[1,2], J. Daniel Diaz[1], Leo A. Kim[1], David M. Wu[1], John B. Miller[1] *

1 Department of Ophthalmology, Retina Service, Massachusetts Eye and Ear, Harvard Medical School, Boston, Massachusetts, United States of America, 2 Department of Ophthalmology, Retina Service, Cornell University Medical School, New York, New York, United States of America

☯ These authors contributed equally to this work.
* john_miller@meei.harvard.edu

**Data Availability Statement:** Anonymized data have been uploaded as a Supporting Information file.

## Abstract

There are multiple surgical approaches to the repair of rhegmatogenous retinal detachment (RRD). Here, we evaluated the outcomes of small-gauge pars plana vitrectomy (PPV), alone or in combination with scleral buckle (SB-PPV), for RRD repair using a standardized technique by 3 vitreoretinal surgeons: "extensive" removal of the vitreous with scleral depression and dynamic examination of the peripheral retina. One hundred eighty seven eyes of 180 consecutive patients treated for primary RRD by three vitreoretinal surgeons at a tertiary academic medical center from September 2015 to March 2018 were analyzed. Most RRDs occurred in males (134 [71.3%] eyes), affected the left eye (102 [54.3%]), and were phakic (119 [63.3%]). PPV alone was performed in 159 eyes (84.6%), with a combined SB-PPV used in the remaining 29 eyes (15.4%); focal endolaser was used in all (100%) cases. The single surgery anatomic success rate was 186 eyes (99.5%) at 3 months, and 187 (100%) at last follow up. Overall best-corrected visual acuity (BCVA) had significantly improved at 3 months ([Snellen 20/47] P<0.00005) and last follow up ([Snellen 20/31] P<0.00005), as compared to day of presentation ([Snellen 20/234]). Our findings suggest that "extensive" removal of the vitreous and dynamic peripheral examination with scleral depression may lead to high single surgery success in primary uncomplicated RRD repair.

## Introduction

Retinal detachment (RD) is a condition defined as separation of the neurosensory retina from the underlying retinal pigment epithelium. Rhegmatogenous retinal detachment (RRD) is the most common type of RD and is characterized by a break in the neurosensory retina with seepage of vitreous fluid into the subretinal space. There are currently several primary interventions employed for repair of RRDs: pneumatic retinopexy (PnR), scleral buckle (SB), and pars

**Funding:** The author(s) received no specific funding for this work.

**Competing interests:** The authors have declared that no competing interests exist.

plana vitrectomy (PPV). Several randomized controlled trials have compared the efficacy of one intervention over another [1–3]. While there is ongoing debate about the optimal modality, PPV is the most performed intervention in the United States for primary RRD repair [4].

Studies in the 1990s reported single surgery anatomic success rates between 74–91% after PPV [5–8]. As surgical instruments and viewing systems improved [9], success of RRD repair also increased to rates of 81–98% with a single surgery [10–18]. Between studies, there was heterogeneity in surgical tools (e.g. viewing systems, illumination) as well as surgical technique. Specifically, the use of adjunct SB, shaving of the vitreous base, utilization of scleral depression, or prophylactic 360˚ endolaser photocoagulation were different based on study and surgeon preference.

Shaving of the vitreous base with scleral depression and dynamic peripheral examination are important in identifying retinal breaks that are not seen preoperatively [19]. Indeed, "extensive", "careful", or "meticulous" shaving of the vitreous and inspection of the peripheral retina has led to high surgical reattachments rates [11, 20–22]. While these terms have no strict surgical definition, they likely represent extensive inspection of the peripheral retina as well as shaving of all possible vitreous at the base, while minimizing the possibility of causing any iatrogenic retinal breaks. This "extensive" removal may lower the risk of formation of new retinal breaks post-operatively by reducing vitreous base traction [23] as well as minimizing the scaffold that may be needed for proliferative vitreoretinopathy (PVR). In the current series, we report the outcomes of RRD repair at a large tertiary academic center following dynamic examination of the far periphery with scleral depression and active flow through the vitreous cutter.

## Methods

### Patient selection

A retrospective review of electronic medical records of consecutive patients who underwent PPV alone or combined PPV with SB for primary RRD between September 2015 and March 2018 was conducted at Massachusetts Eye and Ear (MEE), Boston, Massachusetts. The Partners 2019p000119 Institutional Review Board (IRB) approved the study, which was performed in accordance with the tenets of Declaration of Helsinki and the Health Insurance Portability and Accountability Act (HIPAA). Written informed consent was obtained to perform retinal surgery; however consent to review the surgical outcomes was waived by the IRB as the data were analyzed anonymously. Three (J.B.M., L.A.K., D.M.W.) attending vitreoretinal surgeons performed all vitreoretinal procedures with the assistance of vitreoretinal fellows in training.

Inclusion criteria for the study included eyes with (1) RRD, (2) primary PPV ± SB for RRD, (3) proliferative vitreoretinopathy (PVR) grade B or less [24], and (4) a minimum of 3 months of post-PPV follow-up. Exclusion criteria for the study included any of the following: (1) a history of previous PPV, SB, or combination for RRD, (2) SB surgery alone for RRD, (3) PPV ± SB for RRD performed by vitreoretinal surgeons not affiliated with MEE, (4) insufficient follow-up, (5) PVR grade C or more, (6) RD associated with proliferative diabetic retinopathy, open globe injury, infectious retinitis, and retinoschisis.

### Data collection

Qualitative variables were properly categorized, whereas quantitative data were presented as mean ± standard deviation (SD), or range. Preoperative patient data obtained included date and age at presentation (years), gender, ethnicity, previous ocular surgeries, best-corrected visual acuities (BCVA), reported RD symptoms and their duration, as well as the time interval between presentation and vitrectomy (days). Intraoperative data were also recorded, including

macular detachment, extent of RD, quadrants of the retina involved, type and position of retinal breaks, posterior vitreous detachment (PVD), vitreous hemorrhage (VH), and PVR, and surgical procedures performed. Post-operatively, follow up clinical visits were recorded including BCVA, lens characteristics, and presence of both epiretinal membrane (ERM) and PVR formation. ERM was defined as a hyperreflective change above the inner retinal surface on spectral-domain optical coherence tomography (OCT).

The primary outcome measures were (1) single surgery anatomical success at 3-months, and (2) final reattachment rate at the last follow-up visit. Single surgery anatomical success was defined as attached retina after a single operation without any additional surgical intervention (PPV, SB, pneumatic retinopexy, laser retinopexy or combination) performed for any recurrent RD at 3 months. After 3 months, at last follow up, the other primary outcome measure was defined as final reattachment rate irrespective of either procedural (i.e laser retinopexy) or surgical intervention (e.g PPV).

## Surgical technique

Small (23 and 25 gauge) instrumentation was used for the PPV. Almost all cases utilized the RESIGHT (Zeiss, Jena, Germany) non-contact wide angle visualization system, while a few utilized the BIOM (Oculus Surgical Inc, Port St. Lucie, Florida, USA). All 3 surgeons had equivalent rates of adjuvant SB use. SB was implemented in younger patients (20 of 29 eyes [69%] were less than 60 years old, and 10 eyes [34.5%] were between the ages of 30–50), detachments with inferior breaks (24 out of 85 [28%]), and eyes with PVR [6 out of 24 (25%)].

Extensive 360-degree scleral-depressed shaving of the vitreous base was performed with meticulous inspection of the peripheral retina to identify all pathology. This "extensive" shaving specifically included far peripheral vitrectomy over the ora serrata extending 1–2 mm posterior to the posterior vitreous base with slow moving anterior to posterior scleral depression. All clock hours were carefully shaved and inspected quadrant by quadrant starting at the vertical midline, slightly crossing over, before completing the quadrant at the horizontal axis. A dynamic examination of the far periphery using active flow with the cutter was used to identify additional breaks. At the time of this study, a high cut rate of 7500 cpm and 650 mmHg for vacuum was commonly used to induce elevation of small retinal tear flaps and suspicious lesions within the sphere of influence of the vitreous cutter. In our experience, this helped identify small retinal breaks not seen pre-operatively or with scleral depression alone. In cases with more significant cataracts, posterior capsular opacity (PCO), or capsular phimosis, intravitreal triamcinolone acetonide (TA) was utilized to visualize any potential remaining vitreous. Endolaser photocoagulation was applied in all cases but only to retinal breaks and focal areas of suspicious pathology (e.g. lattice, vitreous tags, hemorrhages, etc.). 360 degree barricade laser was not performed. Flattening of the retina was also accomplished by exchanging fluid for air or injection of perfluorocarbon (PFO).

## Statistical methods

Measured BCVAs were converted to a decimal logarithm of the minimum angle of resolution (logMAR) for subsequent analysis. Acuities of count fingers, hand motions and light perception, were assigned logMAR values of 2.0, 3.0, and 3.3, respectively [22]. Statistical comparison of the continuous data were evaluated by two-tailed Student t-test, with a statistic significance of $P < 0.05$.

## Results

A total of 187 eyes of 180 patients underwent surgical treatment for primary RRD by three VR surgeons: LAK (40/187 of cases), JBM (103/187), and DMW (44/187). The left eye was more commonly involved (54%) (Table 1). One hundred seventy-three (96.1%) patients had a RRD in a single eye, while 6 (3.3%) patients had consecutive (first eye then fellow eye) RDs within the time frame of the study, and 1 (0.6%) patient presented with simultaneous bilateral RRDs. The mean age at the time of diagnosis of RD was 59.6 years (range, 31–86) and 133 patients (71.1%) were male. Most eyes were of Caucasian patients (90.4%), and had no history of previous intraocular surgery (64.7%). One hundred forty-five eyes (77.5%) had ≤2 weeks of symptom duration while 32 eyes (17.1%) had >2 weeks (19 eyes [10.2%] had symptoms between 15–30 days, 6 eyes [3.2%] between 31–60 days, 7 eyes [3.7%] >60 days_ and 10 eyes [5.2%] were unable to provide the duration of symptoms). One hundred and nineteen eyes (63.6%) were phakic, 65 (35.1%) pseudophakic, and 3 (1.6%) were aphakic– 2 eyes of the same patient with Marfan syndrome, who had a remote history of bilateral lens dislocation (left aphakic

**Table 1. Baseline characteristics.**

| | |
|---|---|
| **No. of eyes** | **187** |
| **No. of patients** | **180** |
| **Eye** | |
| Left | 101 (54.0%) |
| Right | 86 (46.0%) |
| **Mean age (range)** | **59.6 (31–86) years** |
| **Gender** | |
| Male | 133 (71.1%) |
| Female | 54 (28.9%) |
| **Race** | |
| White | 169 (90.4%) |
| Non-white | 15 (8.0%) |
| Unknown | 3 (1.6%) |
| **Previous intraocular surgery** | |
| Yes | 66 (35.3%) |
| No | 121 (64.7%) |
| **Duration of symptoms (days)** | |
| ≤14 | 145 (77.5%) |
| 15–30 | 19 (10.2%) |
| 31–60 | 6 (3.2%) |
| >60 | 7 (3.7%) |
| Unknown | 10 (5.4%) |
| **Presenting BCVA** | |
| ≥20/40 | 76 (40.6%) |
| 20/50-20/800 | 60 (32.1%) |
| Count Fingers-Hand Motions | 49 (26.2%) |
| Light Perception | 2 (1.1%) |
| **Lens status** | |
| Phakic | 119 (63.6%) |
| Pseudophakic | 65 (34.8%) |
| Aphakic | 3 (1.6%) |
| **Mean time (std) elapsed from presentation to vitrectomy** | **2.8 ± 3.8 days** |

OU), while the third eye was aphakic since age 2 due to juvenile cataract. In terms of visual acuity, 76 eyes (40.6%) presented with 20/40 or better vision, while 60 (32.1%) were between 20/50 and 20/800, 49 (26.2%) count fingers to hand motions, and 2 eyes (1.1%) were light perception. The mean elapsed time from presentation to surgical intervention was 2.8 ± 3.8 days, with a range of 0–31 days.

Intraoperative findings are shown in Table 2. The macula was found to be detached in 117 eyes (62.6%) and attached in 70 eyes (37.4%). A posterior vitreous detachment was preexisting in 172 eyes (92%), with 15 eyes (8%) requiring surgical induction of a PVD. Vitreous hemorrhage was present in 24 eyes (12.8%). PVR of grade A or B was noted in 24 eyes (12.8%). One hundred and fifteen eyes (61.5%) had multiple breaks, while 69 (36.9%) had a single break; in the remaining 3 (1.1%) eyes, no break was identified. Overall, retinal breaks were slightly more common in the superior retina (53.8%) as compared to 17 eyes (9.2%) with inferior breaks only; 68 eyes (37%) had both superior and inferior retinal breaks. Four eyes (2.1%) had a giant retinal tear. In accordance with location of retinal breaks, 71 eyes (38%) had a superior detachment, 13 (7%) had an inferior detachment, and 103 (55%) had both. The mean number of detached quadrants was 2.3 ± 0.9.

All eyes underwent either a standard 23-gauge (71 [38%]), or 25-gauge PPV (116 [62%]); 158 eyes (84.5%) received PPV alone while 29 eyes (15.5%) received combined PPV with SB (Table 3). Focal endolaser was used only to treat suspected pathology in all eyes (100%). In the 2 of the 3 eyes with no identifiable intra-operative retinal breaks, a scleral buckle was placed, and laser was performed around the drainage retinotomy site as well as the posterior edge of the scleral buckle. In the 3rd eye, additional laser was performed around a previous break that was previously barricaded by laser retinopexy. Intravitreal TA was used in 79 eyes (42.2%). Eleven eyes (5.9%) underwent membrane peel. Drainage of subretinal fluid was achieved most commonly through a preexisting retinal break in 148 eyes (79.1%), followed by drainage retinotomy in 33 eyes (17.7%), and perfluorocarbon (PFO) in 6 eyes (3.2%). Gas tamponade with

**Table 2. Intra-operative characteristics.**

| Macula status | |
|---|---|
| Detached | 117 (62.6%) |
| Attached | 70 (37.4%) |
| Posterior vitreous detachment | 172 (92.0%) |
| Vitreous hemorrhage | 24 (12.8%) |
| Proliferative Vitreoretinopathy | 24 (12.8%) |
| Number of retinal break(s) identified | |
| Single | 69 (36.9%) |
| Multiple | 115 (61.5%) |
| None | 3 (1.6%) |
| Location of retinal break(s) | |
| Superior | 99 (53.8%) |
| Inferior | 17 (9.2%) |
| Both | 68 (37.0%) |
| Giant retinal tear | 4 (2.1%) |
| Location of RD | |
| Superior | 71 (38.0%) |
| Inferior | 13 (7.0%) |
| Both | 103 (55.0%) |
| Mean number of detached quadrants | 2.3 ± 0.9 |

**Table 3. Surgical technique.**

| | |
|---|---|
| **Primary surgical operation** | |
| PPV alone | 158 (84.5%) |
| PPV plus scleral buckle | 29 (15.5%) |
| **PPV gauge** | |
| 25 | 116 (62.0%) |
| 23 | 71 (38.0%) |
| **Endolaser** | 187 (100%) |
| **Intravitreal triamcinolone acetonide** | 79 (42.2%) |
| **Membrane peel** | 11 (5.9%) |
| **Drainage of subretinal fluid** | |
| **Through preexisting retinal break** | 148 (79.1%) |
| **Drainage retinotomy** | 33 (17.7%) |
| **Perfluorocarbon (PFO)** | 6 (3.2%) |
| **Tamponade agent** | |
| Octafluoropropane ($C_3F_8$) | 173 (92.5%) |
| Hexafluoride ($SF_6$) | 11 (5.9%) |
| Silicon oil (SO) | 2 (1.1%) |
| None | 1 (0.5%) |

$C_3F_8$ or $SF_6$ was performed in 173 (92.5%) and 11 (5.9%) eyes, respectively. Silicone oil tamponade was used in 2 eyes (1.1%) due to patient preference (both patients deferred gas because of an occupational need to fly). All eyes had oil removal at last follow up.

This study's main outcome measures centered on single surgery anatomic success rate as well as final reattachment rate at last follow up. At 3 months, 186 out of 187 (99.5%) of eyes were attached following a single surgical procedure (Table 4). At last follow up, all eyes (100%) were attached. The mean follow up period was 511 days ±217.6 days. One hundred fifty-one eyes (80.7%) had at least 1 year of follow up; in the remaining eyes, 20 (10.7%) had at least 6 months of follow up and only 16 (8.6%) had less than 6 months. There were 3 procedural interventions in 3 different patients: case 1 at 1.5 month follow up was noted to have a new

**Table 4. Anatomic and functional outcomes.**

| | |
|---|---|
| Follow-up ± SD, range (days) | 511 ± 217.6 (61–1173) |
| Successful anatomical repair at 3 months | 186 (99.5%) |
| Successful anatomic repair at last follow-up | 187 (100%) |
| **BCVA at last visit** | |
| ≥20/40 | 150 (80.2%) |
| 20/50-20/800 | 35 (18.7%) |
| CF-LP | 2 (1.1%) |
| **Lens status at last visit** | |
| Pseudophakic | 156 (83.4%) |
| Phakic | 28 (13.9%) |
| Aphakic | 3 (1.6%) |
| **Visually Significant ERM** | 15 (8.0%) |
| **Laser for new breaks** | 1 (0.5%) |
| **Laser for new localized RD** | 2 (1.1%) |
| **Recurrent RD requiring PPV/SB/Pneumatic Retinopexy** | 0 |

**Table 5. Visual outcomes.**

|  | Pre-operative BCVA (logMAR) | 3-month BCVA (logMAR) | p-value | Last follow up BCVA (logMAR) | p-value |
|---|---|---|---|---|---|
| All cases | 1.05 | 0.37 | <0.0005 | 0.19 | <0.0005 |
| Macula-on | 0.29 | 0.23 | 0.43 | 0.09 | 0.007 |
| Macula-off | 1.51 | 0.45 | <0.0005 | 0.26 | <0.0005 |

peripheral shallow RD but no clear break that was barricaded with laser retinopexy. Case 2 at 4 months also had a peripheral shallow RD with a new hole and treated with laser barricade. Case 3 at 1-month follow up had an operculated hole discovered elsewhere without associated retinal detachment and underwent supplemental laser retinopexy. All retinae were attached at last follow up. In terms of lens status, 156 eyes (83.4%) were pseudophakic, 28 eyes (13.9%) phakic and 3 remained (1.6%) aphakic. Finally, 15 eyes (8.0%) had developed a visually significant (>20/30) ERM at last follow up.

Post-operative visual outcomes were measured. In terms of BCVA, 150 eyes (80.2%) showed 20/40 or better, while 35 eyes (18.7%) were between 20/50-20/800, and 2 (1.1%) were CF (Table 4). Overall, there was significant improvement in BCVA at 3 months (logMAR 0.37 [Snellen 20/47] P<0.0005) and last follow up (logMAR 0.19 [Snellen 20/31] P<0.0005), as compared to the pre-operative visit (logMAR 1.05 [Snellen 20/234]) (Table 5). Likewise, a similar improvement was seen in the macula-off subgroup at 3 months and last follows up, as well as the macula-on subgroup at last follow up (Table 5, Fig 1). There was no significant change in acuity at 3 months in the macula-on group. We note that the visual acuity analysis does not control for vitreous hemorrhage; in the 12 eyes with VH, 6 (50%) were macula-on, of which several presented with CF vision or worse.

## Discussion

This work evaluated the results of 187 consecutive eyes of 180 patients that underwent surgical repair for primary RRD. The single surgery anatomic success rate was 99.5%, keeping in line with several studies in the literature [1,3,10–12,16–18,21,25,26]. Of the 117 eyes with macula-detached retinal detachment, 116 (99.1%) were reattached with 1 operation, and of the 70 eyes with macula-attached retinal detachment, all (100%) were also reattached with 1 operation. One eye (0.6%) required supplemental laser 1.5 months post-operatively for a localized shallow RD, while a different eye (0.6%) also required focal laser at month 4 for a localized shallow RD; both retinas were attached at last follow up. A third eye (0.6%) was also treated with laser for a newly discovered operculated hole at 1-month follow up. There were no eyes that underwent re-operations (SB, PPV, or PnR) for recurrent RD at last follow up. The final BCVA showed predominantly that 150 eyes [80.2%]) were ≥20/40—no eyes were worse than count fingers. In addition, pseudophakic rates increased from 65 (34.8%) pre-vitrectomy to 156 (83.4%) at last follow up suggesting that cataract surgery also contributed to improved vision.

"Extensive", or "meticulous" shaving of the vitreous has no strict surgical definition to our knowledge. Here, all 3 surgeons used the same "extensive" technique: 360-degree scleral depression to inspect the peripheral retina as well perform vitrectomy over the ora serrata extending 1–2 mm posterior to the posterior vitreous base. Importantly, active flow with the cutter was used to identify additional breaks by inducing elevation of small retinal tear flaps. Additionally, in some eyes, small breaks were identified by the Schlieren effect [27] which was noted either spontaneously or secondary to active flow during vitrectomy.

Shaving of peripheral vitreous releases vitreous traction on the retina and may reduce formation of potential new retinal breaks [28]. High primary anatomic success rates have been

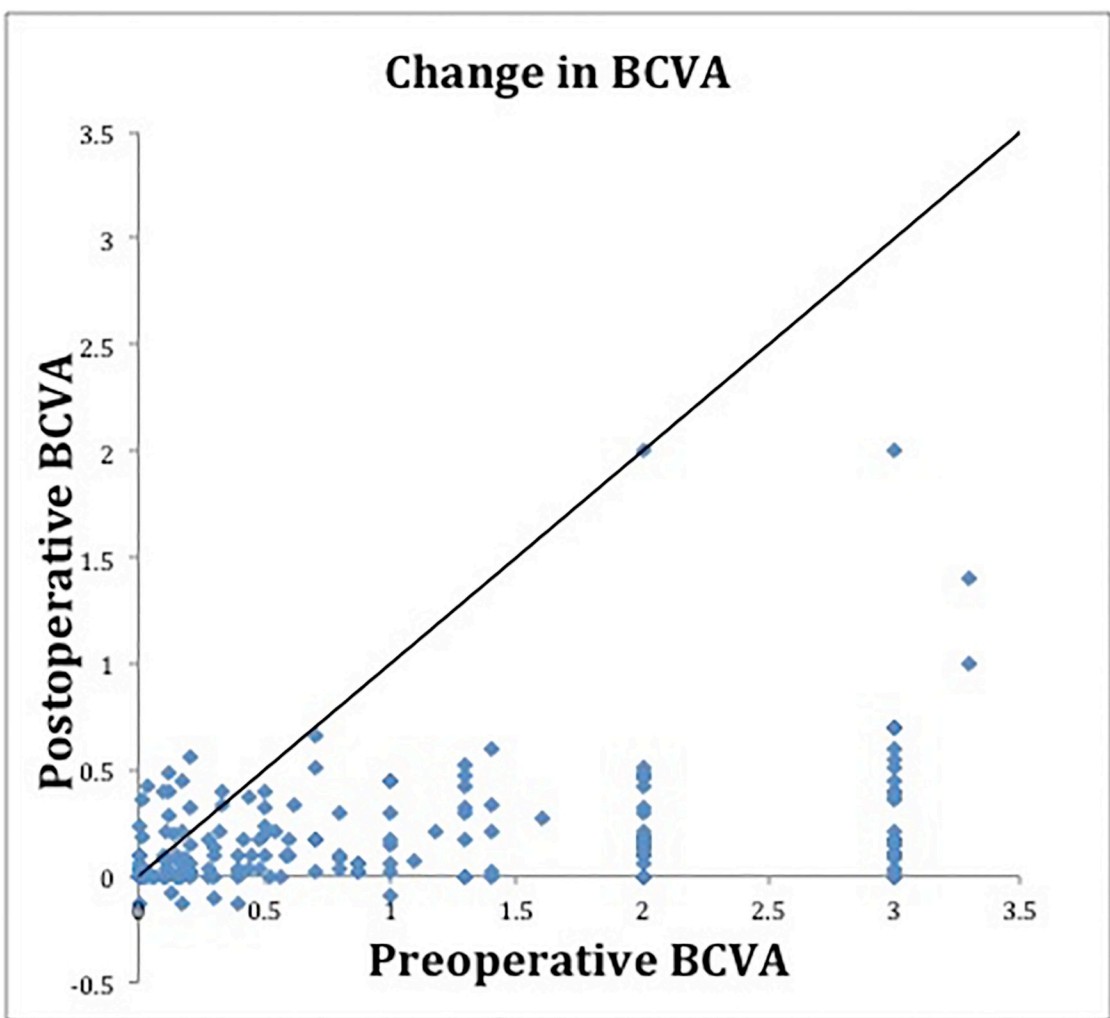

**Fig 1. Change in visual acuity after surgical intervention.** Scattergram showing the best-corrected visual acuity (BCVA, logarithm of the minimum angle of resolution unit) at last follow up as compared to day of presentation.

reported after extensive vitreous removal [10,20,22]. A different study [29] utilized transillumination by light pipe to indent the eye in combination with TA to enhance visualization of the vitreous base for closer vitreous shaving; they reported less detachments after changing to the novel method. In our cohort, if the view into the retina was compromised by a cataract or PCO, then intravitreal TA was used to identify and remove any retained cortical vitreous. At last follow up, only 2 (1.1%) eyes were found to have new retinal holes, with no new breaks in the remaining eyes or clinical evidence of vitreous base traction, suggesting in part thorough vitreous base removal.

At times, we have found that it may be difficult to visualize the peripheral retina up to the ora serrata without scleral depression. Areas in the far periphery with subtle pathologic features may be missed by static examination, endo-illumination or indirect ophthalmoscopy, especially if the retina is lying flat against the RPE. In a retrospective study of primary RRDs without preoperative identification of retinal breaks, 95% of eyes were found to have a break intraoperatively using kinetic indentation of the sclera [30]. A prospective study [31] found 98% of breaks intraoperatively by a similar methodical approach of dynamic examination and scleral depression.

After the discovery of the first break, the study also found multiple breaks in the remaining 30% of retina, underscoring the importance of thorough peripheral dynamic examination.

The current study has several limitations associated with retrospective work. Conclusions that are drawn from the results are limited by the retrospective nature; although the anatomic success rate is high, there are many factors not discussed here (patient population [particularly age or ethnicity] and compliance, clinical diversity, time to surgery, surgeon assistants, etc) that modulate and affect surgical outcomes. Patients with PVR grade C or higher were excluded (though this is in-line with multiple other series). While there was no loss to follow up at 3 months, 16 eyes (8.5%) had less than 6 months of follow up. Thus, while the surgical success rate was high at 3 months and remained high at 1 year, the exact success rate at 1 year may be lower since some eyes received follow up care elsewhere. Several of these patients lived far from our institution and preferred to follow up closer to home–none of them returned to our retina service or emergency room with a new retinal detachment when last reviewed. In addition, triamcinolone was used in ~40% of cases, when the far peripheral view was obscured (e.g. due to cataract, capsular phimosis, or other anterior segment media opacities) in order to identify any residual vitreous for subsequent removal. Although the current work features a large patient population with a long follow-up period, a prospectively designed study is crucial to test the present findings.

## Conclusion

Altogether, our findings suggest that "extensive" removal of the vitreous and dynamic examination of the far periphery with scleral depression and active flow through the vitreous cutter may lead to high single surgery success in primary uncomplicated RRD repair.

This work was presented in part at the Association for Research in Vision and Ophthalmology and the American Society of Retina Specialist annual meetings in 2019.

## Supporting information

**S1 Data.**
(XLSX)

## Author Contributions

**Conceptualization:** Thanos D. Papakostas, Leo A. Kim, David M. Wu, John B. Miller.

**Data curation:** Tedi Begaj, Leo A. Kim, David M. Wu.

**Formal analysis:** Tedi Begaj, Anna Marmalidou, John B. Miller.

**Investigation:** Anna Marmalidou, Thanos D. Papakostas, Leo A. Kim, David M. Wu, John B. Miller.

**Methodology:** Tedi Begaj, Anna Marmalidou, J. Daniel Diaz, John B. Miller.

**Project administration:** John B. Miller.

**Supervision:** John B. Miller.

**Validation:** Anna Marmalidou, Thanos D. Papakostas, John B. Miller.

**Writing – original draft:** Tedi Begaj, John B. Miller.

**Writing – review & editing:** Tedi Begaj, Anna Marmalidou, Thanos D. Papakostas, J. Daniel Diaz, Leo A. Kim, David M. Wu, John B. Miller.

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
