## [Decision Letter · Decision Letter 0]

21 Jul 2020

PONE-D-20-19374

Outcomes of Primary Rhegmatogenous Retinal Detachment Repair with Extensive Scleral-Depressed Vitreous Removal and Dynamic Examination

PLOS ONE

Dear Dr. Miller,

Thank you for submitting your manuscript to PLOS ONE. After careful consideration, we feel that it has merit but does not fully meet PLOS ONE’s publication criteria as it currently stands. Therefore, we invite you to submit a revised version of the manuscript that addresses the points raised during the review process.

Your manuscript has been reviewed by two expert reviewers.  Both reviewer's recommend that revisions are necessary before the manuscript can be further considered for publication. Please carefully read the reviewer's comments and address all questions, concerns and comments in an itemized fashion as described below in a separate letter to the editor. Please also make appropriate changes to the original manuscript using track changes and submit both a final version and a version with track changes so that changes can be easily reviewed. While the reviews are favorable, the paper may not be accepted for publication if the reviewer's comments are not accurately and completely addressed. The revised manuscript may be sent out for further review to the same or different reviewers as well.

We look forward to receiving your revised manuscript.

Kind regards,

Amir H Kashani, M.D. Ph.D.

Academic Editor

PLOS ONE

Reviewers' comments:

Reviewer's Responses to Questions

**Comments to the Author**

1. Is the manuscript technically sound, and do the data support the conclusions?

Reviewer #1: Yes

Reviewer #2: Yes

2. Has the statistical analysis been performed appropriately and rigorously? 

Reviewer #1: Yes

Reviewer #2: Yes

3. Have the authors made all data underlying the findings in their manuscript fully available?

Reviewer #1: Yes

Reviewer #2: Yes

4. Is the manuscript presented in an intelligible fashion and written in standard English?

Reviewer #1: Yes

Reviewer #2: Yes

5. Review Comments to the Author

Reviewer #1: Nice article adding to the body of literature demonstrating the high success rate with vitrectomy and emphasizing the importance extensive shaving practices in achieving high successes. The only criticism would be as the authors alluded to of the high percentage of caucasian's in the cohort.

It would also be nice for the authors to emphasize that while success rate are extremely good at 3 month followup, the authors need to more explicitly state that longer term outcomes > 6 months is unclear due to 8.5% of patients lost to followup.

Reviewer #2: In this paper the authors present Outcomes of over 180 consecutive Primary Rhegmatogenous Retinal Detachment Repairs with Extensive Scleral-Depressed Vitreous Removal and Dynamic Examination.

The paper is well written and the content is appropriate and of interest to vitreoretinal surgeons. Their method described is of interest due to the high anatomic success rate. I have the below comments which may help clarify something of the points raised in the paper:

Just over 15% of eyes had a scleral buckle. The use of scleral buckle and case selection is of interest. Some general comments were made regarding which cases had this procedure done. More specific case selection would be of interest. For example, all patients with inferior pathology or with multiple breaks and inferior pathology or with total detachment and inferior pathology etc had a buckle. Since this is not a small percentage of patients, if the use of buckle was based on one surgeon’s preference or some combination of pathologies, that would be important and useful information to share.

Over 40% of patients had triamcinolone used in the eye to assist in vitrectomy. That is also relatively a high percentage of eyes that had this technique utilized. Was this only used for vitreous shaving based on poor view to the periphery from lens opacity or was it also utilized to elevate the hyaloid or evaluate for the presence of complete posterior vitreous detachment. The criteria of use or whether this was used mostly by one surgeon over others will be useful.

There were a few eyes that were Aphakic at last follow up. Was pars plana lensectomy performed in these eyes during the primary vitrectomy? What was the criteria used to perform the lensectomy.

The active flow used to identify peripheral breaks has not been clearly described in the methods section. This was a technique that the authors credit in their high success rate. A

Better description of how this was used would be helpful. What flow rate or approximate percentage flow rate is used? What is the cut rate of the vitrectomy during this technique. Is that different than the vitreous shaving technique or performed while shaving the vitreous under active scleral depression? What percentage of breaks (in cases of multiple breaks) were identified with this technique (if known)?

All eyes had focal laser performed to identified pathology, yes three eyes had no breaks identified. What was Pathology was identified and lasered in these eyes?

6. PLOS authors have the option to publish the peer review history of their article (what does this mean?). If published, this will include your full peer review and any attached files.

Reviewer #1: No

Reviewer #2: No

---

## [Author Response · Author response to Decision Letter 0]

7 Aug 2020

We thank the editor and reviewer for their hard work. We have responded to each point- it is submitted under the "Response to Reviewers"

---

## [Editor Report · Decision Letter 1]

1 Sep 2020

Outcomes of Primary Rhegmatogenous Retinal Detachment Repair with Extensive Scleral-Depressed Vitreous Removal and Dynamic Examination

PONE-D-20-19374R1

Dear Dr. Miller,

We’re pleased to inform you that your manuscript has been judged scientifically suitable for publication and will be formally accepted for publication once it meets all outstanding technical requirements.

Kind regards,

Amir H Kashani, M.D. Ph.D.

Academic Editor

PLOS ONE

---

## [Editor Report · Acceptance letter]

15 Sep 2020

PONE-D-20-19374R1

Outcomes of Primary Rhegmatogenous Retinal Detachment Repair with Extensive Scleral-Depressed Vitreous Removal and Dynamic Examination

Dear Dr. Miller:

I'm pleased to inform you that your manuscript has been deemed suitable for publication in PLOS ONE. Congratulations! Your manuscript is now with our production department.

Kind regards,

on behalf of

Dr. Amir H Kashani 

Academic Editor

PLOS ONE